# 4-Phenylcoumarin (4-PC) Glucoside from *Exostema caribaeum* as Corrosion Inhibitor in 3% NaCl Saturated with CO_2_ in AISI 1018 Steel: Experimental and Theoretical Study

**DOI:** 10.3390/ijms23063130

**Published:** 2022-03-15

**Authors:** Araceli Espinoza-Vázquez, Francisco Javier Rodríguez-Gómez, Ignacio Alejandro Figueroa-Vargas, Araceli Pérez-Vásquez, Rachel Mata, Alan Miralrio, Ricardo Galván-Martínez, Miguel Castro, Ricardo Orozco-Cruz

**Affiliations:** 1Unidad Anticorrosión, Instituto de Ingeniería, Universidad Veracruzana, Boca del Río, Veracruz 94292, Mexico; rigalvan@uv.mx (R.G.-M.); rorozco@uv.mx (R.O.-C.); 2Departamento de Ingeniería Metalúrgica, Facultad de Química, Universidad Nacional Autónoma de México, Av. Universidad No. 3000, Coyoacán, C.U., Ciudad de México 04510, Mexico; fxavier@unam.mx; 3Instituto de Investigaciones en Materiales, Universidad Nacional Autónoma de México, Ciudad de México 04510, Mexico; iafigueroa@unam.mx; 4Departamento de Farmacia, Facultad de Química, Universidad Nacional Autónoma de México, Av. Universidad No. 3000, Coyoacán, C.U., Ciudad de México 04510, Mexico; perezva@unam.mx (A.P.-V.); rachel@unam.mx (R.M.); 5Tecnologico de Monterrey, Escuela de Ingeniería y Ciencias, Ave. Eugenio Garza Sada 25 01, Monterrey 64849, Mexico; 6Departamento de Física y Química Teórica, DEPg. Facultad de Química, UNAM, Del. Coyoacán, Ciudad de México 04510, Mexico; miguel.castro.m@gmail.com

**Keywords:** AISI 1018 steel, 5-*O*-β-*D*-glucopyranosyl-7-methoxy-3′,4′-dihydroxy-4-phenylcoumarin, inhibitor, EIS

## Abstract

The corrosion inhibition of 5-*O*-β-*D*-glucopyranosyl-7-methoxy-3′,4′-dihydroxy-4-phenylcoumarin (4-PC) in AISI 1018 steel immersed in 3% NaCl + CO_2_ was studied by electrochemical impedance spectroscopy (EIS). The results showed that, at just 10 ppm, 4-PC exerted protection against corrosion with *ղ* = 90% and 97% at 100 rpm. At static conditions, the polarization curves indicated that, at 5 ppm, the inhibitor presented anodic behavior, while at 10 and 50 ppm, there was a cathodic-type inhibitor. The inhibitor adsorption was demonstrated to be chemisorption, according to the Langmuir isotherm for 100 and 500 rpm. By means of SEM–EDS, the corrosion inhibition was demonstrated, as well as the fact that the organic compound was effective for up to 72 h of immersion. At static conditions, dispersion-corrected density functional theory results reveal that the chemical bonds established by the phenyl group of 4-PC are responsible of the chemisorption on the steel surface. According with Fukui reactivity indices, the molecules adsorbed on the metal surface provide a protective cover against nucleophilic and electrophilic attacks, pointing to the corrosion inhibition properties of 4-PC.

## 1. Introduction

AISI 1018 steel is widely used in different industries, especially in construction, despite its low corrosion resistance in various aggressive environments [1]. This steel has excellent mechanical properties and is low-cost [2]. In the oil industry, both general and localized corrosions are the most common types of corrosion appearances. In the flow lines, several problems have been reported, being related to the phenomenon called internal corrosion. In the field of corrosion, there are a few studies regarding the selection and application of organic compounds for protecting metallic surfaces under various conditions; some examples include triazoles [3,4,5,6], benzimidazoles [7,8,9], and quinolones [10,11,12]. These organic molecules, containing heteroatoms, such as oxygen, nitrogen, and sulfur, possessing pairs of free electrons, induce the formation of a film, which delay the process of dissolution of the metal [13]. Nowadays, some plants (Figure 1) have been used to solve this problem in different corrosive media and diverse metallic materials [14,15,16,17,18,19,20,21,22].

The extracts from these plants show effectivity (~60–70%) against corrosion; however, they have been used in high concentrations—higher than those allowed by the official norm “NRF 005 2009”, in accordance with NACE1D-182 [23]. In addition, when using extracts prepared with different solvents, a variety of mixtures of organic compounds are involved; thus, it becomes difficult to determine the true corrosion inhibitor. *Exostema caribaeum* (Jacquin) Roemer and Schultes, commonly known as “copalchi”, is used in traditional Mexican medicine as an antidiabetic and antimalarial agent [24]. This plant belongs to the copalchi complex of the Rubiaceae family, which contains a few species with extremely bitter stembarks. The distinctive secondary metabolites of *E*. *caribaeum* (Figure 2) are 4-phenylcoumarins, with 5-*O*-β-*D*-glucopyranosyl-7-methoxy-3′,4′-dihydroxy-4-phenylcoumarin (4-PC) as the main component. This compound has OH groups with free electrons, which suggests its potential as a corrosion inhibitor agent. The aim of this work was to study the 4-PC corrosion inhibitor in AISI 1018 steel in 3% NaCl + CO_2_ under static and turbulent flow conditions (under which the hydrocarbon flow operate), as well as the persistence of the inhibitor film at the best concentration using the electrochemical techniques.

## 2. Materials and Methods

### 2.1. Isolation and Solution Preparation of the 4-PC from Exostema caribaeum

The isolation of 4-PC was carried out as described in [25,26]. Briefly, an aqueous extract was prepared from the pulverized dry stembark (30 g) using boiling water (5 L) for30 min. The resulting extracts were dried under reduced pressure to yield 10 g of dry extract. A total of 7g of this extract was dissolved in 30 mL of MeOH, and the soluble fraction was subjected to column chromatography on silica gel (100 g), eluting with mixtures of increasing polarity of hexane-ethyl acetate and ethyl acetate-methanol. This process yielded nine fractions (F1–F9). From fraction F4 (106 mg), spontaneously precipitated of 61 mg of 5-*O*-β-*D*-glucopyranosyl-7-methoxy-3′,4′-dihydroxy-4-phenylcoumarin (4-PC), which was identified by spectroscopic and spectrometric procedures.

Next, a dissolution of 0.01 M of 4-PC (p.f. 238 °C, 95% purity) dissolved in ethanol was prepared. Afterwards, the inhibitor was added at different concentrations of 5, 10, 20, and 50 ppm in a corrosive solution of 3% NaCl_,_ saturated with CO_2_ (industrial grade) by bubbling for 0.5 h until reaching a pH of 4.

### 2.2. Electrochemical Evaluation

First, the working electrode (AISI 1018) was previously prepared by polishing with sandpaper from the 320 to 1200 grid. Afterwards, it was mirror-polished with 9 µm alumina. Then, the electrochemical evaluation was carried out using Gill-Ac equipment for open circuit potential (OCP), electrochemical impedance spectroscopy (EIS), and a polarization curve (CP). The working electrode was an AISI 1018 steel using a rotating cylinder (0, 100, and 500 rpm) with an exposed area of 3.92 cm^2^; the reference electrode was Ag/AgCl saturated, and graphite was used as the counter electrode.

#### 2.2.1. OCP vs. Time Change on Chronopotentiograms

This technique was carried out using the electrode cylinder with the area as mentioned above; the potential was stabilized for 800 s in order to reach a stationary state and to perform the electrochemical evaluation.

#### 2.2.2. Electrochemical Impedance Spectroscopy (EIS)

By means of the electrochemical impedance spectroscopy technique, a sinusoidal potential of ±10 mV in a frequency interval (100 KHz to 0.1 Hz) was applied in a three-electrode electrochemical cell.

#### 2.2.3. Immersion Time Effect

By means of EIS, measurements were carried out every 3 h for 3 days using 10 ppm of 4-PC in 3% NaCl + CO_2_ corrosive medium.

#### 2.2.4. Polarization Curves

After the EIS measurements, potentiodynamic polarization curves of the inhibitor at different concentrations were performed, which were measured from −500 to 500 mV in relation to the open circuit potential (OCP), with a speed of 60 mV/min using the ACM Analysis software for data interpretation.

### 2.3. Surface Characterization

Steel samples were immersed for 24 h in either NaCl 3% *w/v* saturated with CO_2_, with and without an inhibitor (4-PC, 50 ppm). The surfaces of the samples were analyzed by scanning electron microscopy (SEM) using a Carl-Zeiss microscope SUPRA 55 VP at 10 kV. The chemical analysis of the resulting corrosive products was performed by energy-dispersive X-ray spectroscopy (EDS) in a timely manner and the time spectrum acquisition is 30%.

### 2.4. Computational Details

The interaction between a single 4-PC molecule and a small iron cluster was theoretically rationalized, by means of dispersion-corrected density functional theory (DFT) methods, to understand the adsorption and corrosion inhibition phenomena on the steel surface. The generalized gradient approximation functional BPW91 [27,28] functional was employed together with the all-electron triple-ζ valence, polarized, and with diffusive functions, Pople’s basis set 6-311++G(2d, 2p) for all atoms [29,30,31,32,33]. Dispersion interactions were considered by means of the Grimme’s “D2” correction term [34,35]. The full method, hereinafter labeled BPW91-D2/6-311++G(2d,2p), was used as coded in the quantum chemistry package Gaussian 09 D.01. Similar methods were used before to study small iron clusters [36,37], organic species, and their compounds [38,39,40,41].

The cluster-based model was assumed by using the Fe_6_-*D_2h_* as the surface. This cluster, in its ground state (GS), exhibits a high spin multiplicity M = 21. To determine the GS structure of the 4-PC molecule, the simplified molecular input line entry system (SMILES) of the 4-PC molecule was used to run a genetic-algorithm (GA)-driven conformational study through the Balloon 1.8.0 software [42,43]. A molecular force field MMFF94 was used to compute the energy of each optimized structure [44]. A total of 300 inequivalent structures were proposed as the initial population, reducing the energy by 20 generations of evolution. Finally, the identified lowest energy structure found was reoptimized at the BPW91-D2/6-311++G(2d,2p) level of theory.

## 3. Results and Discussion

### 3.1. OCP Potential

Figure 3 shows the OCP variation plot. In all cases, the stationary phase was reached after 750 s as a result of a stabilization in redox reactions. These values represent active states on metallic surfaces.

### 3.2. Effect of Concentration and Turbulent Flow

Figure 4 shows the Nyquist diagram of the steel without an inhibitor; at 100 rpm, it reached a Z_re_ value of ~80 Ω × cm^2^. However, at 500 rpm, the semicircle diameter decreased, being attributed to a one-time constant process, related to the charge transference resistance [45].

Figure 5 shows the Nyquist diagrams at different rotation rates and concentrations. In Figure 5a, it is noticeable that the maximum diameter of the semicircle reached a concentration of 50 ppm of 4-PC in static conditions. This can be attributed to the fact that, at higher concentrations, the inhibitor protects the metal surface in the corrosive environment. In these conditions, the Nyquist plots are depressed; we propose the two time constants: charge transference resistance and adsorbed molecules resistance.

Moreover, when rotation rates of 100 and 500 rpm (Figure 5b,c) were evaluated, a better adsorption of the inhibitor was observed, since the Z_re_ value was higher than that for static conditions. This was attributed to the fact that the rotation of the working electrode resulted in a more homogeneous solution, which allowed the inhibitor to reach non-covered sites. On the other hand, the deformation of the semicircles of the Nyquist diagrams shown in Figure 4, called frequency dispersion, can be attributed to different physical phenomena, such as surface roughness, active sites, and lack of homogeneity of solids [46]. In addition, the second time constant observed at low frequencies was due to the organic molecular resistance (Figure 5), which in some cases is not well defined.

Using the equivalent electric circuits of Figure 6, the electrochemical parameters were determined (Table 1) for the different hydrodynamic conditions studied. The equivalent electric circuit of Figure 6a is used when the inhibitor is not added to the system. Figure 6b corresponds to the addition of different concentrations of the inhibitor.

**Figure 6 ijms-23-03130-f006:**
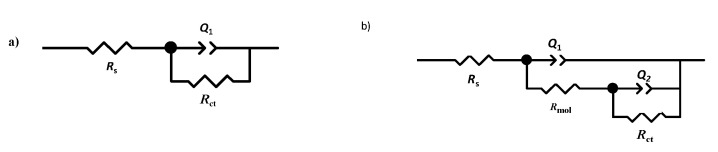
Equivalent electric circuits used in the system (**a**) without an inhibitor and (**b**) with inhibitor.

where *R*_s_ is the solution resistance, *R*_ct_ is the charge transfer resistance, *R*_mol_ is molecule resistance, and *Q* is the constant phase element (CPE).

CPE is used instead of a pure condenser for the deviation of the ideal dielectric behavior related to the lack of homogeneity in the electrode surface. The impedance (Z) of the CPE is given by the following expression:(1a)$$Z_{CPE}=Q^{-1}{\left(j\omega \right)}^{-n}$$

$C_{\mathrm{dl}}$ represents a double layer capacitance (Equation (1b)):(1b)$$C_{\mathrm{dl}}=Y_{0}{\left(\omega_{m}^{\prime \prime }\right)}^{n-1}$$
where Q represents the value of CPE, n is the exponent of CPE (which can be used as an indicator of heterogeneity or roughness in the surface), and $\omega_{m}^{\prime \prime }$ is the angular frequency in rad/s.

Depending on n, CPE can represent a resistance (Z_CPE_ = R, n = 0), a capacitance (Z_CPE_ = C, n = 1), a Warburg impedance (Z_CPE_ = W, n = 0.5), or an inductance (Z_CPE_ = L, n = -1).

The inhibition efficiency (*ƞ*) [47] is given by Equation (2):(2)$$\eta \;\left(\%\right)=\frac{{\left(\frac{1}{Rp}\right)}_{Blank}-{\left(\frac{1}{Rp}\right)}_{inhibitor}}{{\left(\frac{1}{Rp}\right)}_{Blank}}\;x100$$
where *R*_p_ is the polarization resistance with and without an inhibitor (*R*_p_ = *R*_ct_ + *R*_mol_)

Figure 7 shows some examples of the adjustment of experimental data, calculated using a concentration of 10 ppm at different rotation rates. A good adjustment (fit linear) was observed with the equivalent electric circuit values shown in Figure 6b.

Table 1 shows the electrochemical parameters of 4-PC at different rotation rates: at static conditions (0 rpm), the *R*_p_ value did not increase as the concentration did. When the rotation rate was 100 rpm, an important increment in this parameter was observed. This was attributed to the fact that the inhibitor reached the metal surface faster, protecting the metal surface from the corrosion process. Finally, the same trend of effects was observed at 500 rpm using a concentration of 4-PC of 10 ppm, showing the best inhibition efficiency (*ղ*~94%).

On the other hand, the *C*_dl_ values decreased as the concentration of the organic compound increased. The variation in *R*_ct_ and *C*_dl_ values can be correlated with the gradual movement of the water molecules over the surface of the electrode in the presence of 4-PC, leading to a decrease of active sites, delaying the corrosion process [48,49].

**Table 1 ijms-23-03130-t001:** Electrochemical parameters of 4-PC as a corrosion inhibitor in 1018 steel at different rotation rates, immersed in 3% NaCl + CO_2_.

Rotation Rate (rpm)	*C* (ppm)	*R*_s_ (Ω cm^2^)	*n*	*C*_dl_ (μF cm^−2^)	*R*_p_ (Ω cm^2^)	*ղ* (%)	±SD
	0	0.9	0.8	351.0	83.0	-	-
0	5	27.0	0.7	687.0	321.4	72.7	6.4
	10	26.6	0.8	359.9	282.3	68.3	8.5
	20	33.1	0.8	391.9	241.7	64.9	5.0
	50	27.5	0.9	343.9	337.8	75.4	0.3
	0	0.8	0.8	435.3	30.0	0.0	0.0
100	5	8.0	0.8	377.6	558.0	94.6	0.2
	10	10.6	0.9	245.6	823.1	96.0	1.2
	20	9.4	1.0	176.9	517.8	93.5	2.1
	50	13.1	0.9	165.9	968.6	96.4	1.3
	0	3.3	0.9	785.1	36.0	0.0	0.0
500	5	17.0	0.7	333.4	428.8	91.1	2.1
	10	14.2	0.7	123.7	764.6	94.0	2.8
	20	20.8	0.9	113.9	406.1	90.3	2.9
	50	6.6	0.9	43.3	416.2	91.3	0.1

### 3.3. Effect of Immersion Time

Another important parameter to establish the useful life of a corrosion inhibitor is the immersion time [50,51], which, measuring the effective period of time (days, hours), it protects a metallic surface. The Nyquist diagrams at three different times (Appendix A) revealed that, in all cases, the Z_re_ values diminished as the time increased. These diagrams showed two time constants—one related to the charge transference resistance and the other linked to the organic molecules. It is important to note that the interaction of the inhibitor with the metallic surface decreased in strength as the inhibition efficiency did; consequently, a process of desorption of the organic molecules took place. This behavior was further observed when the variation of *η* as a function of the immersion time was plotted (Figure 8). From this curve, it was evident that an acceptable protection against corrosion of up to 18 h of immersion was reached.

### 3.4. Polarization Curves

The polarization curves of the 1018 steel, immersed in 3% NaCl + CO_2_, with and without an inhibitor, are shown in Figure 9. It is clear that the corrosion current density (icorr) increased in the presence of the inhibitor, attributed to the fact that it delayed the metal dissolution process.

On the other hand, when the concentration increased, both anodic and cathodic curves moved towards the lower current density. This phenomenon implies that the inhibitor might suppress the anodic reaction of the metal dissolution, as well as the detachment of cathodic hydrogen [52]. Finally, at a concentration of 5 ppm, the 4-PC behaved as an anodic inhibitor, while at 10 and 50 ppm, the behavior was cathodic.

### 3.5. Adsorption Process

The adsorption isotherms can provide important information about the interaction of the inhibitor with the metallic surface, among which, two processes stand out: physisorption and chemisorption [53]. Physisorption takes place between the positive active centers of the electrons of the benzene rings with the metallic surface, while chemisorption is due to the formation of coordination bonds between the molecules of an inhibitor and the d orbitals of iron atoms over the steel surface [54].

It is possible to find several adsorption models that describe the process in which an inhibitor can be adsorbed in the metallic surface; the most important are the following:(3)$$\frac{C}{\theta }=\frac{1}{kads}+C\;\mathrm{Langmuir}\;\mathrm{model}\;$$
(4)$$C^{1/n}K_{ads}=\theta \;\mathrm{Freundlich}\;\mathrm{model}\;$$
(5)$$Ck_{ads}=\left(\frac{\theta }{1-\theta }\right)e^{f\theta }\;\mathrm{Frumkin}\;\mathrm{model}\;$$

Analysis of the data using the three models (Equations (3)–(5)) revealed that the best adjustment was attained with the Langmuir model (Figure 10), presenting a linear behavior that accords well to the value of the correlation coefficient.

After analyzing the best isotherm that describes the behavior of the 4-PC, the Gibbs standard energy of adsorption was calculated [22]:(6)$$\Delta G{{}^{\circ}}_{ads}=-RT\ln k_{ads}$$
where *R* is the constant of ideal gases, *T* is the absolute temperature, and *k*_ads_ is the equilibrium constant.

Table 2 shows the thermodynamic parameters obtained. It is important to note that the Δ*G*°_ads_ is higher than −40 KJ/mol at 100 and 500 rpm, which implies a chemisorption process. At static conditions (0 rpm) the adsorption process resulted from a combination of physisorption and chemisorption.

### 3.6. Surface Morphology

SEM images were obtained with and without the best concentrations of 4-PC (10 ppm) being immersed in 3% of NaCl + CO_2_ to complete the EIS study. As shown in Figure 11a, in the presence of the 4-PC, the steel surface was completely polished without any damage. A damaged was observed when the organic compound was not added (Figure 11c), showing several corrosion products on the surface. Finally, in the presence of 4-PC (Figure 11b), the metal surface was less damaged, attributed to a protective film of this inhibitor in the metal surface.

Finally, the chemical analysis of each sample (Figure 12) showed that there was a presence of a corrosive species of chlorine and oxygen in the metal immersed in the corrosive medium (Figure 12b). However, the metal treated with 4-PC did not contain the corrosive species of chlorine and the concentration of oxygen species decreased significantly. These findings might be related to the formation of a 4-PC protective film in the active sites.

### 3.7. Adsorption Models

Firstly, Figure 13a shows the GS structure of the 4-PC molecule. It is appreciated that the β-*D*-glucopyranosyl appears almost parallel, aligned to the substituted phenyl group, whereas the rest of the 4-phenylcoumarin is obtained perpendicular to both. As explained in the computational details subsection, the optimized GS structures of 4-PC molecule and Fe_6_-*D_2h_* iron cluster were used to set the initial structures of the metal corrosion inhibitor compound. Twelve inequivalent initial structures, with the molecules separated by 1.4 Å as the shortest interatomic distance, were used to scan the potential energy surfaces (PES) with multiplicities M = 21, 19, and 17. According with its point group, *D*_2h_, there are only three different iron–iron bonds in Fe_6_ iron cluster and then three adsorption sites: face-centered, atop, and bridge. Thus. the potential energy surfaces (PES) of the Fe_6_ + 4-PC compound, with multiplicities M = 21, 19, and 17, were studied; the most feasible structures are shown in Appendix A. Only local energy minima on the PES were considered, as indicated by their all-real vibrational frequencies. All possible adsorption modes were characterized by the zero-point energy (ZPE) corrected, with energies relative to the GS (Δ*E*_ZPE_).

ZPE-corrected relative energies, to the GS, were calculated as stability indicators of the Fe_6_ + 4-PC compound. This relative energy was calculated as follows: Δ*E*_ZPE_ = *E*_ZPE_(Fe_6_ + 4-PC) − [*E*_ZPE_(Fe_6_) + *E*_ZPE_(4-PC)], where terms refer to the ZPE-corrected energies of the Fe_6_ + 4-PC, Fe_6_, and 4-PC systems, respectively. Among all structures optimized, Appendix A only illustrates the ones obtained with favorable, negative, ZPE-corrected relative energy values.

Mostly, feasible structures of Fe_6_ + 4-PC show the 4-PC moiety adsorbed on the metal cluster by the 4-phenylcoumarin (Appendix A). Conversely, glucopyranosyl and methoxy groups are not appealing to be attached to the metal surface. The GS structure, the one with the most negative Δ*E*_ZPE_ relative energy, is characterized deeply throughout this manuscript (see Figure 13a).

On the other hand, free binding energy, *G*_Bind_, was computed to compare it directly with the Gibbs standard energy of adsorption Δ*G*°_ads_, experimentally determined for static conditions. *G*_Bind_ was calculated as follows: *G*_Bind_ = *G*(Fe_6_ + 4-PC) − [*G*(Fe_6_) + *G*(4-PC)], where terms refer to the free energies of the Fe_6_ + 4-PC, Fe_6_, and 4-PC systems at 300 K, respectively. Appendix A shows the structures with *G*_Bind_ obtained as negative values, as functions of being considered as spontaneous adsorption at 300 K. In this context, the structure with the most negative *G*_bind_ coincide with that with the most negative relative energy. The *G*_Bind_ value obtained (−37.99 kJ/mol) for the GS structure agrees admirably well with the Δ*G*°_ads_, of about −37.34 kJ/mol, experimentally obtained for static conditions. Moreover, both values agree with the borderline chemisorption and mixed chemisorption-physisorption regime.

In its ground state, the substituted phenyl group present in the 4-PC molecule is mainly responsible of the adsorption of the organic corrosion inhibitor on the Fe_6_ iron cluster. An important elongation of carbon–carbon bond lengths is appreciated, from 1.396 to 1.462 Å. Iron–iron interatomic distances are enlarged above 2.2 Å, whereas the shortest iron–carbon bond established accounts 2.165 Å. Conversely, hydroxyl groups close to the iron cluster remain almost unchanged.

### 3.8. Global Parameters

Now, it is necessary to rationalize the reasons why 4-PC is able to interact with the metal surface. Consequently, global parameters and several isosurfaces are proposed to elucidate the interactions and adsorption process of the 4-PC molecule on the steel surface. Energies and isosurfaces of the highest occupied molecular orbital (HOMO) and the lowest unoccupied molecular orbital (LUMO) were obtained (see Table 3 and Figure 14). The higher in energy *E*_HOMO_, the easier the sharing of electrons to the iron surface, leading to a more effective corrosion inhibitor. In the current case, 4-PC obtained an *E*_HOMO_ value (−5.532 eV) considerably higher (less negative) in comparison with other corrosion inhibitors recently reported, such as fluconazole (*E*_HOMO_ = −6.516 eV). The HOMO–LUMO gap, *E*_gap_, obtained for the 4-PC molecule (2.764 eV) is considerably higher than that obtained for the isolated Fe_6_ iron cluster (0.438 eV). Thus, the naked iron cluster is expected to be more reactive than 4-PC, similarly, the resultant compound Fe_6_ + 4-PC. In contrast, *E*_gap_ of 4-PC is considerably smaller than those obtained in case of amino acids (3.7757–4.413 eV) [54], xanthenes (3.21–5.27 eV) [55], benzaldehyde thiosemicarbazone derivatives (3.411–3.8633 eV) [56], pyrimidine derivatives (4.062–4.613 eV) [57], alkyl carboxylates (4.839–7.557 eV) [58], and even the recently studied fluconazole 4.424 eV, to mention a few, pointing to its higher reactivity [59]. Despite the above, the analysis of the spatial distribution of HOMO and LUMO frontier orbitals, detailed below, denotes how the highly reactive metal surface is passivated by the 4-PC molecule.

Deep into the inhibitor–metal interactions, the global descriptors described within Pearson’s hard-soft acid-base (HSAB) theory were calculated as well (see Table 3). Calculated values for vertical ionization potential (*I* = 7.313 eV) and electron affinity (*A* = −0.835 eV) of 4-PC molecule (see Table 3) were employed to calculate the global hardness, *η* = (*I* − *A*)/2, global electronegativity, *χ* = (*I* + *A*)/2, as well as global electrophilicity, *ω* = (*I* + *A*)^2^/(8*(*I*−*A*)) (see Table 3); moreover, the fraction of electrons transferred Δ*N* = (*χ*_bulk-Fe_ − *χ*_4-PC_)/[2*(*η*_bulk-Fe_ + *η*_4-PC_)] [59], where the values assumed are: *χ*_bulk-Fe_ = 7 eV and *η*_bulk-Fe_ = 0 eV for bulk iron [60].

The hardness exhibited by 4-PC molecule (4.074 eV) is slightly lower than that obtained in case of fluconazole (4.197 eV) [40] and alkyl carboxylates (4.082–4.749 eV) [58]. Thus, invoking the HSAB principle, a softer inhibitor could interact easier with soft acids, just like the iron metal surface (*η*_bulk-Fe_ = 0 eV) or the Fe_6_ iron cluster (2.361 eV). However, other corrosion inhibitors are softer than 4-PC, for instance mycophenolic acid (3.773 eV) [41], xanthenes derivatives (1.61–2.63 eV) [60], pyrimidine derivatives (3.749–4.158 eV) [57]), and amino acids (1.89–2.21 eV) [54].

Global electronegativity, χ of 4-PC of about *χ* = 3.239 eV, is obtained pretty close to the 3.861 eV obtained in case of Fe_6_ iron cluster. Thus, both species have similar tendencies to attract electrons from the surrounding. Electrophilicity indices ω calculated for the 4-PC molecule is considerably lower, calculated as *ω* = 0.644 eV, than that obtained for the Fe_6_ iron cluster, of about *ω* = 1.578 eV. Consequently, the tendency to donate electrons from the inhibitor molecule to the metal surface can be illustrated by this index. Similarly, the positive fraction of electrons transferred, Δ*N*, relative to the corrosion inhibitor molecule and the bulk–iron surface, calculated as 0.462, is another index pointing to the charge transference from the organic molecule to the metal surface [39]. These predictions, based on HSAB global parameters, were confirmed by a population analysis, detailed below.

### 3.9. Frontier Orbitals, Charge Transference, and Electrostatic Potential Maps

In the current case, both HOMO and LUMO frontier orbitals are mostly located at the substituted 4-phenylcoumarin moiety. In particular, oxygen atoms show contributions of their non-bonding p orbitals. Moreover, carbon rings exhibit π-bonding orbitals delocalized on the whole 4-phenylcoumarin. Thus, regions likely to donate, as well as accept electrons, are those surrounding the 4-phenylcoumarin moiety. The above explains why the substituted phenyl group of 4-PC molecule adsorbs on the metal surface in several low-lying structures (see Appendix A). Moreover, HOMO and LUMO, alpha and beta, frontier orbitals of the composed system Fe_6_ + 4-PC exhibit σ-bonding orbitals established by the overlap between carbon p orbitals of the phenyl group and iron d orbitals coming from the iron cluster. Thus, a prominent chemisorption component of the adsorption process of the 4-PC molecule on the steel surface can be visualized by the multiple bonds formed between them. Moreover, HOMOs and LUMOs show mostly contributions from overlapped iron d orbitals, leading mostly to π- and δ-bonding orbitals.

The most prominent chemisorption component was already explained, but the physisorption components also of interest. Consequently, the electrostatic contribution to the Langmuir-type physisorption–chemisorption process between 4-PC molecule and the steel surface will be explained by electrostatic potential (ESP) maps and a natural bond orbital (NBO) population analysis. Thus, the ESP mapped on van der Waals surfaces and NBO charge distributions for these systems were obtained (see Figure 13b).

The ESP map calculated for the isolated 4-PC molecule reveals that regions with the lowest electrostatic potential, consequently those with excess electrons, correspond to the oxygen belonging the hydroxyl groups. Moreover, the same oxygen atoms exhibit the most negative NBO charges. Conversely, regions with the highest ESP and the most positive NBO charges are precisely those around hydrogen atoms belonging to hydroxyl groups. Thus, it is possible to suppose that, at the beginning of the adsorption process, long range electrostatic interactions could be established by the polar groups, hydroxyl ones, and the metal surface. However, the charge transference from the 4-PC molecule to the iron cluster takes place only to the iron atoms closest to the inhibitor molecule, ranging from −0.227 to −0.157 e (see Figure 13b). Since chemisorption is the most important contribution to the adsorption process, it is expected that the charge transferred by 4-PC will be shared with the surface iron atoms on steel. Thus, the electrostatic interaction between the corrosion inhibitor and metal surface, after the adsorption, is rather limited.

Finally, the passivation of the metal surface will be understood in terms of Fukui reactivity indices. First of all, the most prominent sites for nucleophilic and electrophilic attacks would be characterized by the condensed Fukui functions, *f*^+^ and *f*^−^, respectively. These indices were calculated, for a given atom A, as follows: *f*^+^(A) = q_N_(A) − q_N+1_(A) and *f*^−^(A) = q_N__−1_(A) − q_N_(A), where q_N_(A) is the total NBO charge calculated for the neutral system at the GS structure and with N total electrons. According with both indices, shown in Figure 13c, iron atoms are the most suitable sites for nucleophilic—as well as electrophilic—attacks. Conversely, the 4-PC molecule shows the lowest values for both reactivity indices. Therefore, the organic molecule provides an almost inert cover to both kinds of attacks. The above is consistent with the approximation to the Fukui function by means of frontier orbitals, since HOMO_α,β_ and LUMO_α,β_ isosurfaces resembles the correspondent for *f*^+^ and *f*^–^ functions, respectively. Overall, the 4-PC molecules, mostly chemisorbed by their phenyl groups on the metal surface active sites, provide a less reactive and, consequently, protective cover against corrosion on the AISI 1018 steel surface.

## 4. Conclusions

In summary, a rather good effectivity against corrosion of the 4-PC was observed. This behavior was also detected even in turbulent flow conditions (100 or 500 rpm) at only 10 ppm when analyzed using EIS. The polarization curves demonstrated that 4-PC behaved as a mixed type of inhibitor, being adsorbed in the metallic surface, according to the Langmuir model. These results were corroborated with the SEM–EDS analysis. The chemisorption contribution to the adsorption process, at static conditions, was explained by the chemical bonds established between the phenyl group of 4-PC and the metal atoms. Electrostatic interactions, caused by polar groups of the corrosion inhibitor, are limited. Condensed Fukui indices reveal that the cover formed by inhibitor molecules provides a less reactive and, consequently, effective barrier against nucleophilic and electrophilic attacks.

## Figures and Tables

**Figure 1 ijms-23-03130-f001:**
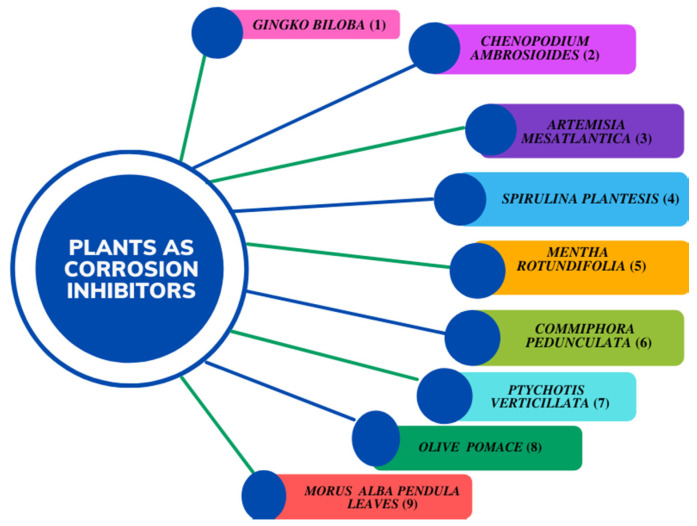
Plants that have been used as corrosion inhibitors: (1) [13,14], (2) [15], (3) [16], (4) [17], (5) [18], (6) [19], (7) [20], (8) [21] and (9) [22].

**Figure 2 ijms-23-03130-f002:**
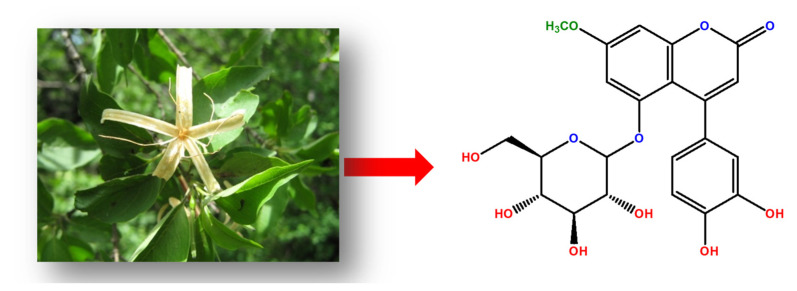
*Exostema caribaeum* and chemical structure of 4-PC.

**Figure 3 ijms-23-03130-f003:**
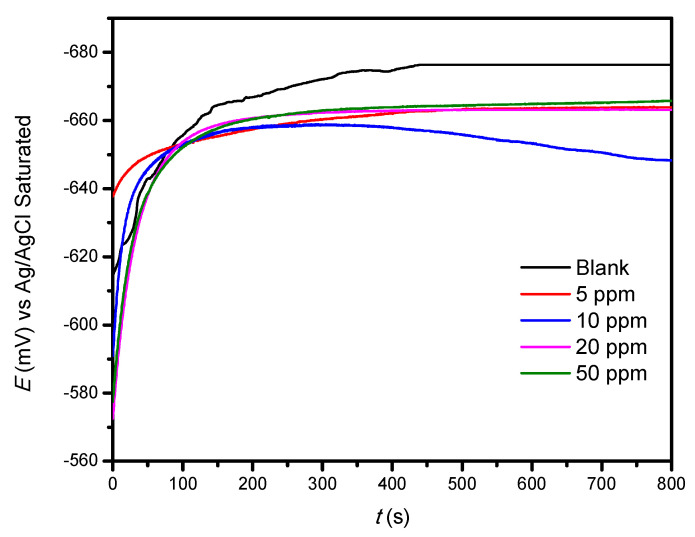
Potential variations at different concentrations of 4-PC as corrosion inhibitors in 1018 steel immersed in 3% NaCl + CO_2_.

**Figure 4 ijms-23-03130-f004:**
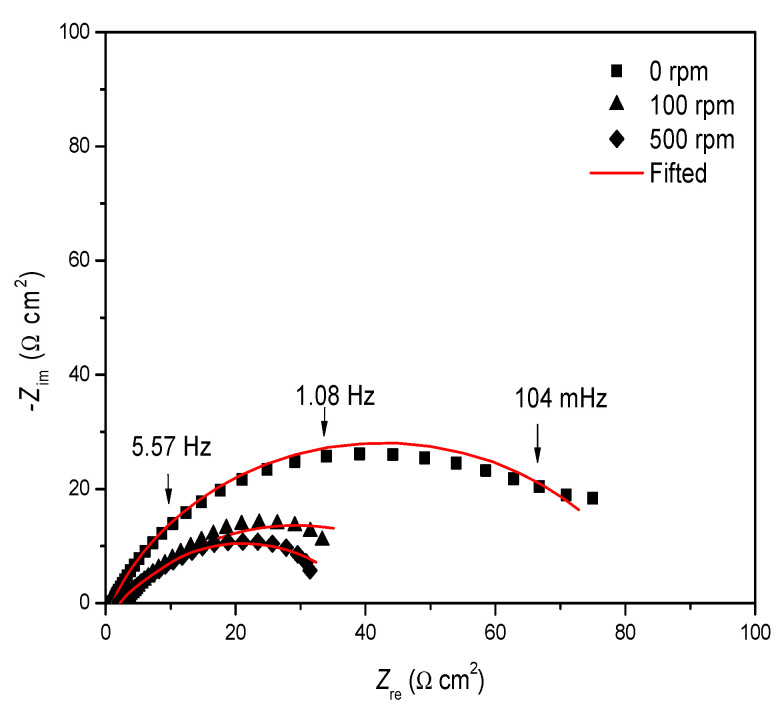
Nyquist diagram of 1018 steel immersed in 3% of NaCl + CO_2_ at different rotation rates.

**Figure 5 ijms-23-03130-f005:**
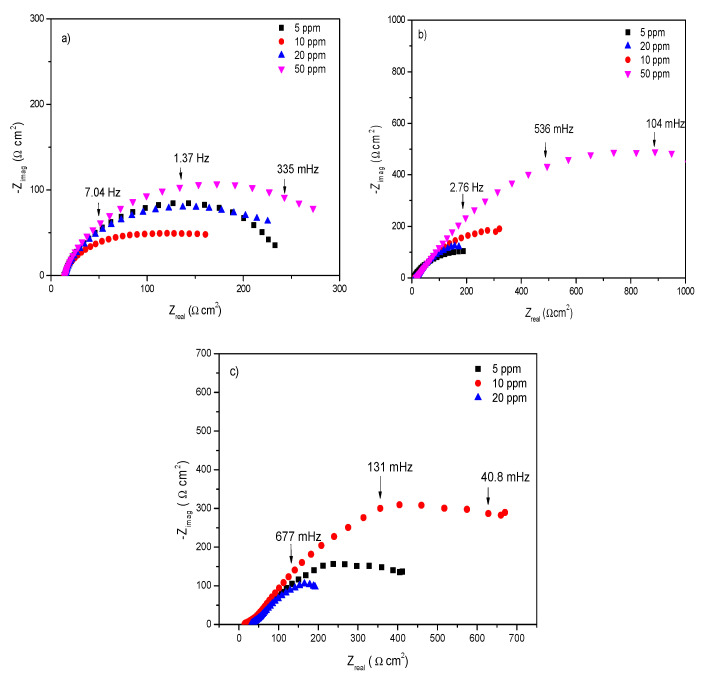
Nyquist diagram of 4-PC in 1018 steel immersed in 3% of NaCl + CO_2_ at different rotation rates (**a**) 0 rpm, (**b**) 100 rpm and (**c**) 500 rpm.

**Figure 7 ijms-23-03130-f007:**
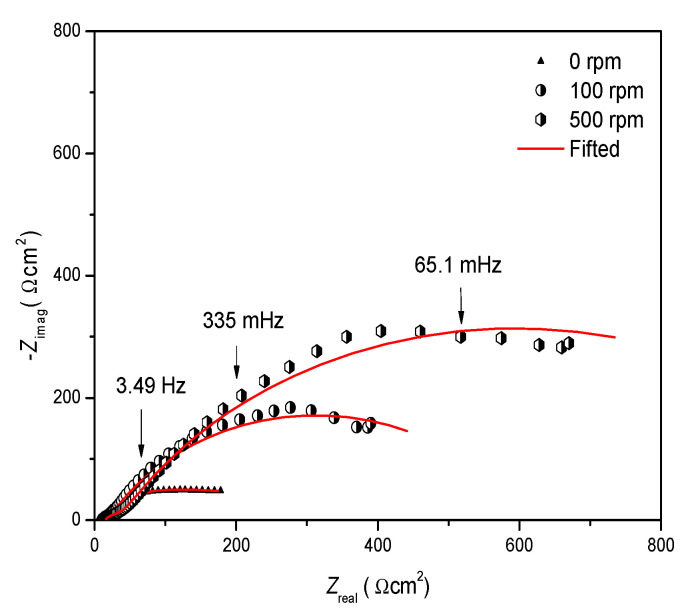
Examples of Nyquist plots of 10 ppm of 4-PC at different rotation rates.

**Figure 8 ijms-23-03130-f008:**
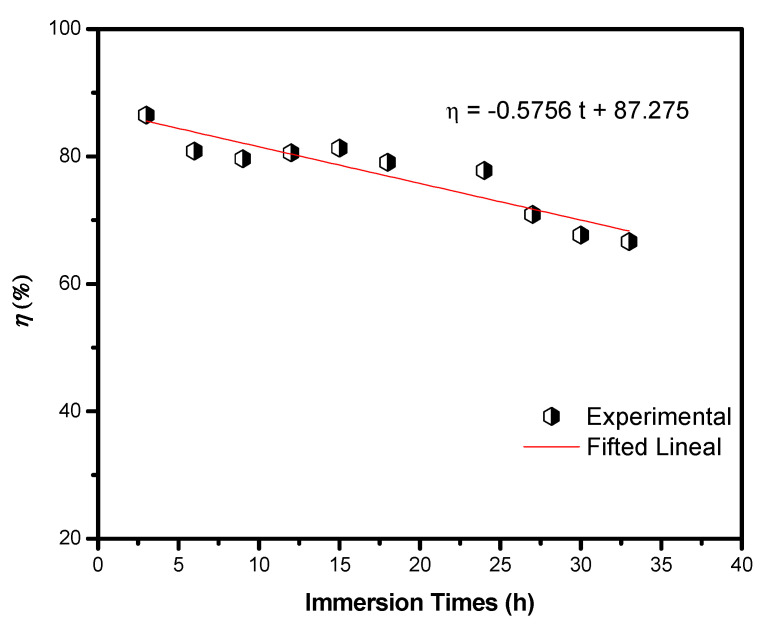
Variation of inhibition efficiency of 4-PC in function of the immersion time, in 3% NaCl + CO_2_ for AISI 1018 steel.

**Figure 9 ijms-23-03130-f009:**
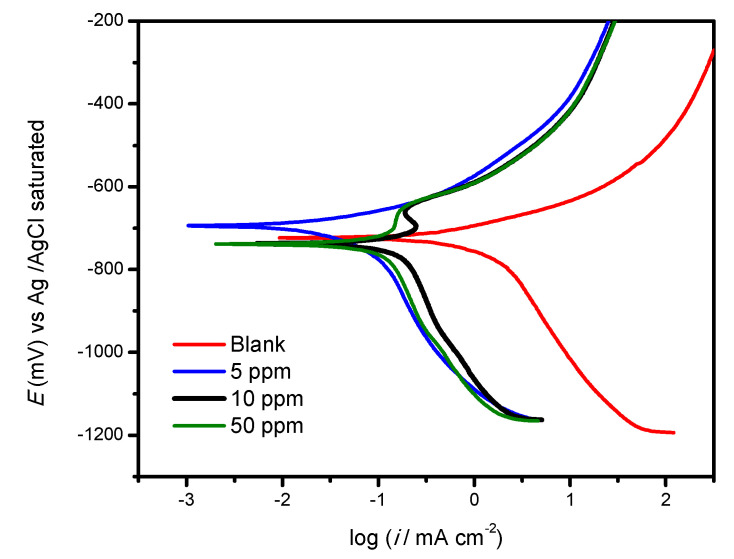
Polarization curves of 4-PC in AISI 1018 steel immersed in 3% NaCl + CO_2_.

**Figure 10 ijms-23-03130-f010:**
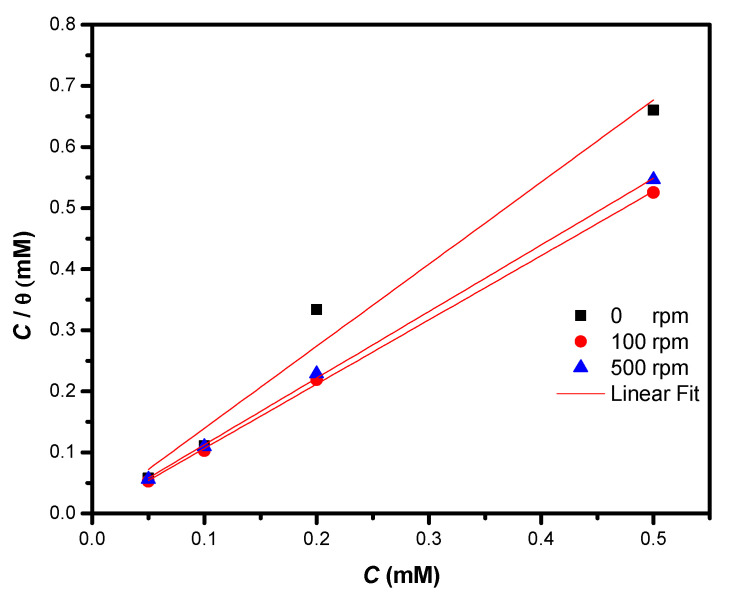
Adsorption isotherm of 4-PC immersed in 3% NaCl + CO_2_ in 1018 steel at different rotation rates.

**Figure 11 ijms-23-03130-f011:**
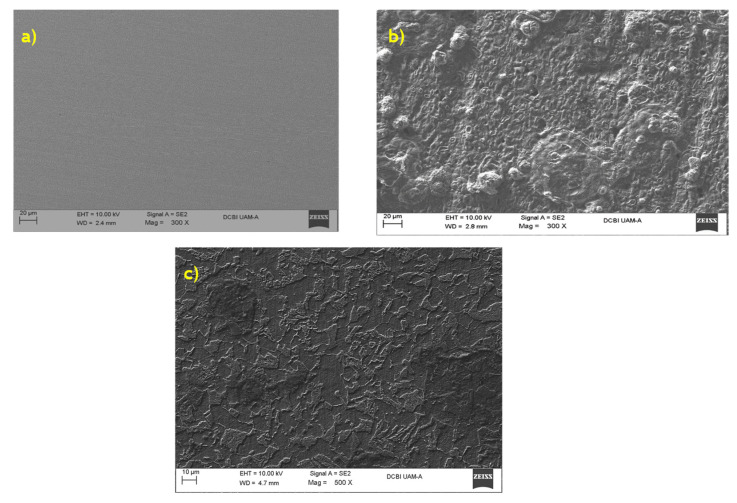
SEM images at 300 X of (**a**) AISI 1018 steel with no-inhibitor; (**b**) AISI 1018 steel immersed in NaCl + CO_2_; (**c**) 10 ppm 4-PC+ NaCl + CO_2_ after 24 h.

**Figure 12 ijms-23-03130-f012:**
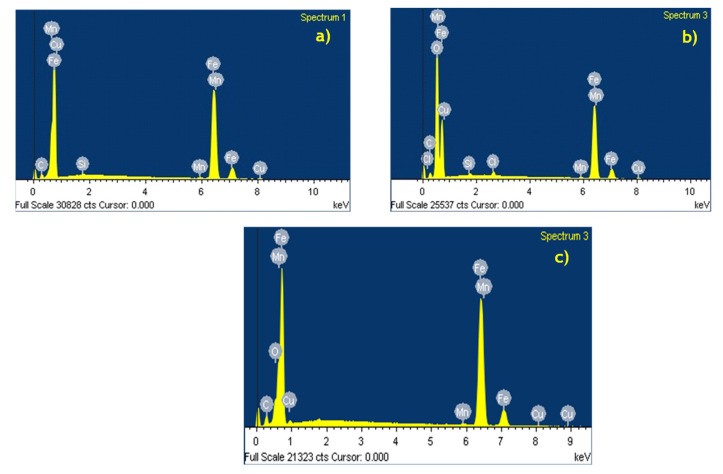
EDX images at 300 X of (**a**) AISI 1018 steel with no-inhibitor; (**b**) AISI 1018 steel immersed in NaCl + CO_2_; (**c**) 10 ppm 4-PC + NaCl + CO_2_ after 24 h.

**Figure 13 ijms-23-03130-f013:**
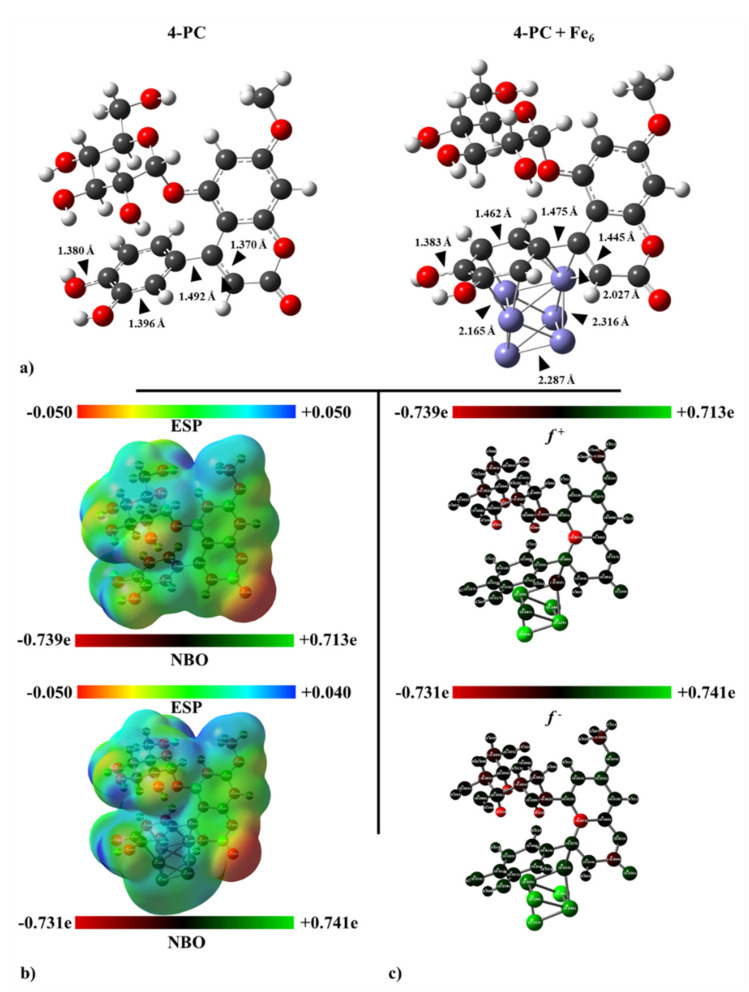
(**a**) Optimized structures and relevant bond lengths of the 4-PC molecule, isolated and interacting with Fe_6_. (**b**) NBO charge distribution and ESP, mapped on isosurfaces with 0.0004 a.u. of the electron density, of the 4-PC molecule, isolated and interacting with Fe_6_. (**c**) Condensed Fukui indices for nucleophilic and electrophilic attacks, *f*^+^ and *f*^−^, calculated for the 4-PC molecule, isolated and interacting with Fe_6_, calculated by means of NBO charges. Results obtained at the BPW91-D2/6-311++G(2d, 2p) level.

**Figure 14 ijms-23-03130-f014:**
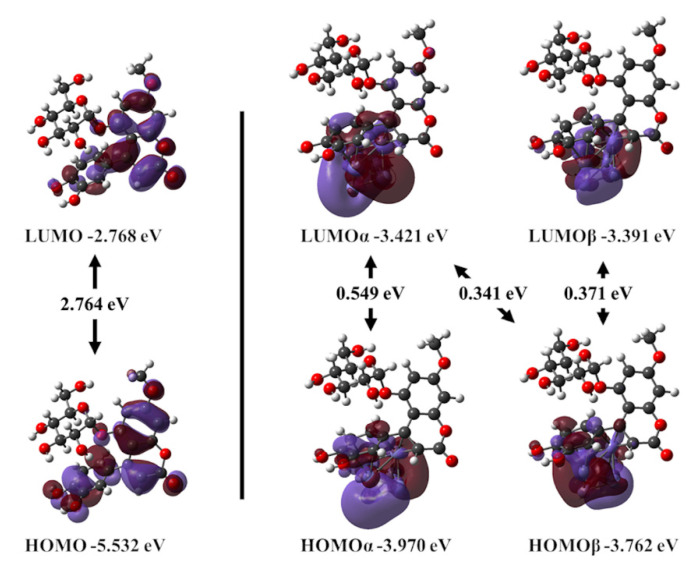
Frontier orbitals HOMO and LUMO of the 4-PC molecule (**left**) and Fe_6_ + 4-PC (**right**), plotted on isosurfaces with 0.02 a.u. of the electron density. HOMO and LUMO, alpha and beta, calculated at the BPW91-D2/6-311++G(2d,2p) level.

**Table 2 ijms-23-03130-t002:** Thermodynamic analysis of 4-PC immersed in 3% NaCl + CO_2_ in 1018 steel at different rotation rates by means of the Langmuir model.

Rotation rate (rpm)	*Δ*G*° _ads_* (KJ mol^−1^)	Linear Regression Equation (M)	R^2^
0	−37.34	*C*/*Ɵ* = 1.3058 *C* + 4 × 10^−6^	0.9782
100	−43.22	*C*/*Ɵ* = 1.0522 *C* + 3 × 10^−7^	0.9995
500	−41.00	*C*/*Ɵ* = 1.0902 *C* + 8 × 10^−7^	0.9995

**Table 3 ijms-23-03130-t003:** Energetic and global parameters calculated for 4-PC, Fe_6_ cluster, and their compounds. Values between parentheses are related to beta, spin down, orbitals.

System	*E*_HOMO_ (eV)	*E*_LUMO_ (eV)	*E*_gap_ (eV)	*I* (eV)	*A* (eV)	*η* (eV)	*χ* (eV)	*ω* (eV)	Δ*N*
4-PC	−5.532	−2.768	2.764	7.313	−0.835	4.074	3.239	0.644	0.462
Fe_6_	−4.333 (−3.969)	−2.369 (−3.531)	0.438	6.222	1.500	2.361	3.861	1.58	
Fe_6_ + 4-PC	−3.970 (−3.762)	−3.421 (−3.391)	0.341						

## Data Availability

Not applicable.

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
