# Peer review of "4-Phenylcoumarin (4-PC) Glucoside from *Exostema caribaeum* as Corrosion Inhibitor in 3% NaCl Saturated with CO_2_ in AISI 1018 Steel: Experimental and Theoretical Study"

_ijms, 2022, doi:10.3390/ijms23063130_

Round 1
Reviewer 1 Report
1 Introduction
- since there are no further references in the text to figures 1 a-i, it is necessary to remove the letters in the captions to the figures.
- it is necessary to expand the introduction. Write in more detail why AISI 108 steel was used in the study.
- more clearly formulate the purpose of the study.
2. Presentation of the material
The article lacks a part describing the methodology, which makes it difficult to read and understand the material presented in this paper. One gets the impression that the authors mixed up sections 2 and 3. Usually, the experimental part is located after the introduction. Thus, the information must be given in the following form:
1 Introduction
2. Materials and Methods
3. Results and Discussion
4.Conclusions
3. Section 3
3.2. Preparation of solutions
Make a single part regarding the research solution used and inhibitory additives. Moreover, the authors tried to combine these sections: «A dissolution 0.01 M of 4-PC (p.f. 238°C, 95% purity) dissolved in ethanol was prepared. Afterwards, the inhibitor was added at different concentration of 5, 10, 20 and 50 ppm in the corrosive solution of 3% p/v of NaCl. saturated with CO2 (industrial grade)». Give a transcript of 4-PC; correct "3% p/v of NaCl. saturated with CO2 (industrial grade)" with 3% NaCl. It is not entirely clear how the solution was saturated with CO2? If the solution was purged with CO2, then this should be written. What instrument was used for conducting electrochemical studies? It also needs to be written about. Also, please provide information about sample preparation. If the sample preparation was carried out according to the sentence: «The working electrode was previously prepared by polishing with sandpaper from 320 to 1200 grid. Afterwards, it was mirror polished with 9 μ alumina», then move this sentence to this section.
3.3.1. OCP vs time change on Chronopotentiograms
In this section you must add information about the size of the steel samples, their shape, relative to which electrode the data were obtained.
3.3.2.1 Immersion time effect
"By means of EIS, measurements were carried out every 3 hours for 3 days using 10 ppm of 4-PC in 3% NaCl + CO2 corrosive medium" it is necessary to explain why this immersion mode was chosen. If these tests were carried out according to the regulatory documents, then it should be referred to.
Section 3.4. Surface characterization
"The chemical analysis of the resulting corrosive products was performed by energy-dispersive X-ray spectroscopy (EDS). It is necessary to specify from which area the EDS was made (in a point or from the whole surface of the sample) and the time of the spectrum acquisition.
3.5 Computational details
There is a lot of information in this section, it should be shortened.
In the methodical section information about the study of adsorption of the inhibitor should be added.
According to the authors' results, this section should be placed before section 3.4. I propose to leave part of it in the methodology and the rest in the results. In the methodology: "The adsorption isotherms can provide important information about the interaction of the inhibitor with the metallic surface, among which, two processes stand out: Physisorption and chemisorption [39]. Physisorption takes place between the positive active centers of the electrons of the benzene rings with the metallic surface, while chemisorption is due to the formation of coordination bonds between the molecules of an inhibitor and the d orbitals of iron atoms over the steel surface [40]. It is possible to find several adsorption models which describe the process in which an inhibitor can be adsorbed in the metallic surface; the most important are the following:
equation of Langmuir model (3)
equation of Freundlich model (4)
equation of Frumkin model (5)».
4. Rework the figures
Figure 3 - usually give figure from cathode potential to anode potential, not the other way around. Please correct.
Figure 11 - need to make the scale marker readable
Figures 13, 15, and 16 - place figures next to each other, not one below the other.
5. The text should be checked for typos and inaccuracies
For example: ZCPE (CPE lined font); change [36-37] on [36,37]; correct 3% NaCl + CO2 (CO2 lined font); Fi -gure 1 must be written together.
Reviewer 2 Report
The manuscript is concerned with the use of 4-PC as a corrosion inhibitor.
The manuscript includes experimental research and discussions, as well as theoretical considerations. The manuscript is interesting, well prepared. the studies presented are well planned and mostly sufficiently discussed.
The manuscript contains only minor errors, and in several places the lack of consistency in the notations used can be noticed.
Sometimes symbols are written with or without subscripts (for example: the in Figure 6 and lines 114-133 (Rp - Rp; ZCPE - ZCPE; the symbols with subscript in Table 1.
Also 2.6. Adsorption models (EZPE - EZPE; GBind - GBind) ; EHOMO, ELUMO;). Please compare with the symbols in 3. Materials and methods.
Other comments were taken below:
- line 5, affiliation; please change the order (current order: 1,3,2,…).
- line 21, abstract; CO2 without subscript.
- line 41; "Fi-gure 1", please correct.
- line 52; please add reference to the NOM NRF 2009 standard. Is there no relevant international standard?
- 2.1. OCP Potential; the discussion of OCP is brief and concise, can the discussion of OCP be developed.
- line 88; "the inhibitor is degraded in the corrosive environment", please justify your statement, on what basis for such conclusions?
- Figure 5; please add explanations of a, b and c in the description.
- Equation 2; in the context of previous comments, I propose to analyze the equation (the symbols 1/Rp and subscripts for blank and inhibitor).
- description of figure 7, line 139; I understand that it is a system with 6b, the "B" requires at least some explanation.
- figure 9; curve for 10 ppm hardly visible.
- tables 2; "+ CO2" is missing from the description.
- figures 11 and 12; the description, "NaCl + CO2" without spaces, please correct.
- 2.6. Adsorption models; claster Fe6 without subscript, in the other part of the manuscript the Authors use notations with subscript, please standardize. Additionally, line 255, "Fe6+ 4-PC".
- line 417, 3.2. Preparation of solutions; there is no space between the value and the degree symbol.
Round 2
Reviewer 1 Report
Colleagues, thanks for the work done, however, there are still things that need to be corrected before the paper can be published.
1. Introduction
Check the numbering of the references throughout the text. There are now references 1-13 in the introduction, and then there are references 23, etc. Where did references 14 through 22 go? This needs to be fixed.
2. Materials and methods.
Correction of section numbering:
- Section 3.2. Electrochemical evaluation should be corrected to section 2.2. Electrochemical Evaluation
- Section 2.3.2 Electrochemical Impedance Spectroscopy (EIS) should be corrected to section 2.2.2 Electrochemical Impedance Spectroscopy (EIS). In the same section check the end of the sentence: "was applied in a three-electrode electrochemical cell according to section 2.2". At this time there is no section 2.2 in this article. What section are the authors referring to? Please correct accordingly.
-Section 2.3.2.1. Immersion time effect should be corrected to section 2.2.3. Immersion Time Effect
-Section 2.3.3. Polarization Curves should be corrected to section 2.2.4.
-Section 2.4. Surface Characterization should be corrected to section 2.3. Surface Characterization
-Section 2.5. Computational details should be corrected to section 2.4. Computational Details
3. Results and Discussion
Correction of numbering of sections:
- Section 2.4. Polarization curves should be corrected to section 3.4. Polarization Curves
- section 3.4. Adsorption process should be corrected to section 3.5. Adsorption Process
- section 2.5. Surface morphology should be corrected to section 3.6. Surface Morphology
- section 3.6. Adsorption models should be corrected to section 3.7. Adsorption models
- section 3.7. Global parameters should be corrected to section 3.8. Global Parameters
- section 3.8. Frontier orbitals, charge transference and electrostatic potential maps should be corrected to section 3.9. Frontier orbitals, charge transference and electrostatic potential maps. 4.
4. Figures
Figure 11a - please replace the gray color with white (the field under the microphoto).
Figure 12 - make the spectra bigger. Right now the chemical elements are not well readable.
